# The Simulation Design of Microwave Absorption Performance for the Multi-Layered Carbon-Based Nanocomposites Using Intelligent Optimization Algorithm

**DOI:** 10.3390/nano11081951

**Published:** 2021-07-29

**Authors:** Danfeng Zhang, Congai Han, Haiyan Zhang, Bi Zeng, Yun Zheng, Junyao Shen, Qibai Wu, Guoxun Zeng

**Affiliations:** 1School of Computers, Guangdong University of Technology, Guangzhou 510006, China; dfzhang@gdut.edu.cn; 2School of Materials and Energy, Guangdong University of Technology, Guangzhou 510006, China; hca0109gdut@163.com (C.H.); Junyao-Shen@gdut.edu.cn (J.S.); wuqb@gdut.edu.cn (Q.W.); zenggx@gdut.edu.cn (G.Z.); 3State Key Laboratory of Oncology in South China, Sun Yat-sen University Cancer Center, Guangzhou 510060, China

**Keywords:** microwave absorption, electromagnetic parameters, reflectivity, genetic algorithm, artificial bee colony algorithm

## Abstract

The optimal design objectives of the microwave absorbing (MA) materials are high absorption, wide bandwidth, light weight and thin thickness. However, it is difficult for single-layer MA materials to meet all of these requirements. Constructing multi-layer structure absorbing coating is an important means to improve performance of MA materials. The carbon-based nanocomposites are excellent MA materials. In this paper, genetic algorithm (GA) and artificial bee colony algorithm (ABC) are used to optimize the design of multi-layer materials. We selected ten kinds of materials to construct the multi-layer absorbing material and optimize the performance. Two algorithms were applied to optimize the two-layer MA material with a total thickness of 3 mm, and it was found that the optimal bandwidth was 8.12 GHz and reflectivity was −53.4 dB. When three layers of MA material with the same thickness are optimized, the ultra-wide bandwidth was 10.6 GHz and ultra-high reflectivity was −84.86 dB. The bandwidth and reflectivity of the optimized material are better than the single-layer material without optimization. Comparing the GA and the ABC algorithm, the ABC algorithm can obtain the optimal solution in the shortest time and highest efficiency. At present, no such results have been reported.

## 1. Introduction

Microwave absorbing coatings were first used in military stealth technology. Stealth technology can effectively improve the survival and penetration ability of weapons and equipment, showing great power in modern war. A stealth fighter is coated with a layer or multi-layer absorbing material in the fuselage for avoiding radar tracking. Microwave absorbing coatings are not only widely used in military fields, but also increasingly important in civil fields such as electromagnetic compatibility and microwave radiation protection. Microwave equipment, communication transmitting stations, power transmission and transformation equipment and mobile phones all have electromagnetic radiation. Electromagnetic interference produced by electromagnetic radiation not only affects the realization of high performance for electronic products, but also causes long-term or short-term harm to the human body. Therefore, it is possible to use microwave absorption coatings for microwave radiation protection. The research and development of MA materials has important application value [1,2,3,4]. The MA materials are required to absorb electromagnetic waves as high as possible within a given frequency range. Therefore, the materials are required to have a wide absorption bandwidth, strong absorption, light weight, and thin thickness [5,6,7,8]. 

According to the loss mechanism, MA materials can be divided into magnetic loss type and dielectric loss type. Magnetic loss materials include ferrite, carbonyl iron, ferromagnetic metal, etc. [9,10]. Ferromagnetic metal micro-powder microwave absorbing materials are composed of ferromagnetic metal iron, cobalt, nickel and their alloys, which loses electromagnetic wave through magnetic hysteresis loss, eddy current loss, etc. Compared with ferrite series MA materials, ferromagnetic metal powder has higher permeability and saturation magnetization, better magnetic loss and temperature stability [11,12]. However, the application of ferromagnetic micro-powder is limited by its narrow absorbing bandwidth, high density, easy oxidation and poor corrosion resistance. Nowadays, ferromagnetic metal micro-powder materials are modified mainly through coating and doping [9,13]. Dielectric loss absorbing materials such as silicon carbide, carbon materials (carbon fiber, carbon nanotubes, porous carbon), etc., whose mechanism is dielectric polarization relaxation loss. Compared with metal materials, carbon materials are widely used in field of MA due to their light weight, high dielectric constant and good chemical stability [14,15,16]. Khurram et al. studied the honeycomb core material filled with 10 wt.% carbon powder and obtained a reflectivity of −7 dB with a bandwidth of 18 GHz [17]. However, although carbon materials have wide-band and excellent dielectric loss, its weak magnetic property leads to poor magnetic loss. 

In recent years, a number of new carbon-based MA nanomaterials (carbon nanotubes, carbon coated metal particles and graphene) have provided a nanoscale MA composite material [18,19,20,21,22]. Lu et al. [20] studied the simulated reflectivity of carbon nanotube composites with shell-core structure up to −47 dB and the bandwidth of −20 dB up to 2.0 GHz; Wan et al. [23] studied the composite material with Fe–Co alloy particles grown on the surface of carbon fiber. When the thickness was 1.8 mm, they had a minimum reflectivity of −37.7 dB; Huang et al. [24] studied carbon-coated nickel nanoparticles, when the coating thickness was 3 mm, minimum reflection loss was −39.8 dB and absorbing band width was 8.4 GHz; Zhang et al. [25] prepared nanocomposite particles with CoFe_2_O_4_ as the core and carbon nanotubes (CNT) as the shell. They found that the CNT grew on the surface of CoFe_2_O_4_ microspheres. When the coating thickness was 2 mm, the maximum reflection loss was −32.8 dB, and the absorption bandwidth was 5.7 GHz. Sun et al. [26] prepared a FeNi/graphene nanocomposite material combining dielectric loss and magnetic loss. When the thickness of the absorbing coating was 1.5 mm, it had a minimum reflect loss of −32 dB at 12.4 GHz and a bandwidth of 3.3 GHz. Liu et al. [27] obtained a good impedance match on porous carbon coated nickel nano-absorbing material. When the thickness was 2.6 mm, the minimum reflectivity was −51.8 dB. Chan et al. [16] used carbon nanotube/ferromagnetic metal micro-powder composite absorbing materials in antenna design to eliminate various parasitic signals and improve antenna performance. As one of the most promising new MA materials, nano carbon-based absorbing composite materials will likely obtain high MA performance. In this paper, based on these excellent absorbing materials (such as carbon nanotubes, carbon coated iron nanoparticles, nickel nanoparticles, micron iron powder, etc.), the performance optimization design of micro-nano multilayer composite absorbing materials will be studied.

At present, the research of MA materials is mainly toward the direction of low-dimensionality, complexity, and multi-functionality and compatibility. Compatibility requires that the absorbing material has a wide-band absorbing characteristic. The complexity of materials means using composites of multiple materials to meet the comprehensive performance of MA materials. However, these requirements are mutually restricted. For example, it is easy to design a kind of absorbing material with thin thickness, but its absorbing bandwidth and strength are insufficient, we can also easily design a high absorption material within a certain amount of bandwidth, but it is too heavy or too thick to achieve the demand of the application. Therefore, it is very important to combine various materials and design a MA material with good comprehensive performance. To design the MA materials with thin thickness, light weight, wide bandwidth and high absorption performance is a typical optimization problem. Compared with single layer absorbing coating, multi-layer coatings can obtain better performance by adjusting material type and thickness of each layer. Thickness of each layer, electromagnetic parameters of absorbing material and material order will all affect the absorbing properties. Compared with the single-layer absorbing coating, the multi-layer absorbing coating is more beneficial to widen the absorption bandwidth and enhance the reflection loss [28,29,30].

There are many factors influencing the reflectivity of the MA material. The combination of the optimal dielectric materials is mostly empirical at present. When there are many material layers or too many types of materials, the optimization design is time-consuming and laborious using experimental method only, and it needs computer-aided optimization design. When using multiple dielectric materials to match the design of multilayer absorbing materials, the relationship between the combined absorbing performance and their original absorbing performance of each material is not a simple linear relationship, so that the results of the electromagnetic optimization are highly nonlinear, multi-parameter and multi-peak features, and the use of algorithms will greatly improve the efficiency of optimization design.

At present, the genetic algorithm (GA) has been used in the optimization design of MA materials. Wang [31] optimized the performance of absorbing coatings by using GA algorithm, and the reflectivity in the specified frequency interval achieves the best comparing no optimization. The calculation results show that each layer thickness of the material, the type of material and the arrangement order of the materials, the design results are accurate and effective, and the use of GA greatly improves the efficiency of optimization design. Compared with other algorithms, the GA optimization has higher accuracy, but a vast calculating amount. Feng [32] optimized the reflectivity of the absorbing material with an improved GA. The two obtained layers of absorbing material (inner layer 0.9 mm, outer layer 2.7 mm) can realize the reflectivity less than −15 dB within a wide range of 8–18 GHz. Jiang [33] presented the application of the NSGA-II for constructing Pareto optimal designs of microwave absorbers. Results indicate that the NSGA-II can work more efficiently than traditional Pareto genetic algorithms. N. Dib [34] presented the optimal design of multilayer microwave absorbers using differential evolution (DE) with competitive control-parameter setting technique, and a very wideband (0.1–20 GHz), thin (total thickness of 6.8 mm) seven-layer absorber has been designed. Artificial bee colony (ABC) algorithm, as a new swarm intelligence optimization algorithm, has become another important optimization algorithm in the field of biomimetic intelligence computing. Toktas [35,36,37] designed multilayer radar absorbing material by utilizing a predefined material set including electrical variables existing in the literature, and various numbers of layers are optimally determined using artificial bee colony (ABC). They presented a three-dimensional objective space optimization strategy using an enhanced multi-objective artificial bee colony (ABC) algorithm for the design optimization of layered radar absorbing material, and optimized material are picked up from a composite material database with 51specimens from nine previously reported studies. Yigit [38] optimized the thicknesses, sequence and number of layers of the MRA structures using the materials given in the literature by triple-objective artificial bee colony algorithm optimizations at the frequency ranges of 2–18 GHz for each angle of incidence from 0° to 60°. However, until now, there is no report about the application of ABC algorithm for the optimization design of multilayer new carbon-based MA micro-nano composite materials. In our work, we selected optimizing bandwidth and reflectivity as the objective function for design, and apply ABC and GA algorithm to give the optimal assembling of several new carbon-based absorbing materials, and the purpose of optimizing the absorbing bandwidth and reflectivity performance is achieved. At the same time, we discuss the efficiency of different optimization algorithms for the same problem based on different algorithms mechanisms. According to the theory of electromagnetic wave transmission line, matlab software is used to simulate reflectivity curve after optimization [21].

## 2. Experimental

### 2.1. Preparation of Material and Electromagnetic Parameter for Testing Samples

Commercially available carbon nanotubes with a diameter of 20–40 nm are raw materials. Carbon nanotubes/paraffin composites with carbon nanotubes mass fraction of 7 wt.% and 9 wt.% were made into ring-shaped solid samples ($\varphi_{in}=3.04\;mm$,$\;\varphi_{out}=7.00\;mm$). The electromagnetic parameters of samples were measured by a vector network analyzer in the frequency range of 2–18 GHz.

Ni@C nanoparticles were prepared by arc discharge method. The arc discharge was generated by applying a direct current of 150 A at 60 V between two electrodes at an argon pressure of 10 kPa. The distance between the electrodes was 3–4 mm. Micron-sized (μm) graphite and iron powder were used as raw material. The mixture was shaped in form of cylindrical anode of 25 mm in diameter and 50 mm in height. This anode was consumed and produced soot during the arc discharge process. This soot was deposited on the inner surface of the reaction chamber. After the arc discharge reaction, the soot was collected and the samples were obtained. The ring samples were prepared according to the mass fraction of 40, 50, and 60 wt.% Ni@C nanoparticles. The preparation process of Fe@C nanoparticles was the same as the Ni@C nanoparticles. The ring samples of Fe@C were prepared according to the mass fraction of 40 and 50 wt.%.

Commercially available iron powder with a particle size less than 58 μm was selected as raw material and was reduced by the ball grinding process. The grinding ball and iron powder were weighed in the mass ratio of 20:1, and grinded for 8 h and 10 h, respectively, at the grinding speed of 400 rpm. The particle sizes of iron powder for 8 h and 10 h were between 10–20 μm and 5–15 μm, respectively. The samples were labeled as Fe-8 h and Fe-10 h, respectively. An iron powder/paraffin ring sample was prepared according to the mass fraction of 50 wt.% iron powder for the measurement of electromagnetic parameters. 

### 2.2. The Optimization Design of Microwave Absorption Performance of Multi-Layer Absorbing Coating Using Algorithm

The GA and ABC algorithm was used to optimize the performance of 10 kinds of absorbing materials (Table 1). Considering the requirements of practical application, we choose: (1) 2–3 layers of absorbing material, and the material of each layer is one of the materials in Table 1. The total thickness of absorbing coating will be controlled within 3 mm, and the minimum thickness of each layer will not be less than 0.5 mm. (2) The designed frequency band is 2–18 GHz. (3) The optimal bandwidth (F1) and the optimal reflectivity (F2) are selected as the objective function for the optimization design. Through the optimization of the algorithm, the type and order of the matched material, the corresponding thickness of each layer of material, the bandwidth, and the minimum reflectivity can be obtained.

#### 2.2.1. Design of Setting Parameters for Optimizing the Performance of Multi-Layer Absorbing Coating Using GA 

For optimal operation of multi-layer MA materials using GA, the flowchart of GA algorithm operation is show in Figure 1. The operating parameters should be set in advance: The population size (the number of individuals contained in the population) was set as 50. Chromosome length was set as 13 for the two-layer absorbing material and 22 for the three-layer absorbing material. The termination evolutionary algebra of genetic algorithm was set as 5000. The crossover probability was set as 0.6. The variation probability was set as 0.01. Two objective functions are set as:

For the design of optimal bandwidth, the objective function F1 is set as:(1)$$F1={\left|f_{i}\right|}_{RL=-10\;}-f_{i+1}{}_{\left|RL=-10\right|}|\;\;(i=1,\;3\ldots )$$
where ${\left|f_{i}\right|}_{RL=-10\;}$ denotes the frequency when the reflectivity reaches −10 dB.

For the design of optimal reflectivity, the objective function F2 is set as:(2)$$F2=min\left(RL\left(\theta ,f_{i}\right)\right)\;(i=1,\;2\ldots )$$
where $RL\left(\theta ,f_{i}\right)$ denotes the reflectivity of all frequency points in the design band at various angles of incidence.

#### 2.2.2. Design of Setting Parameters for Optimizing the Performance of Multi-Layer Absorbing Coating Using ABC Algorithm 

When ABC algorithm is used to optimize the multi-layer MA materials, the flowchart of ABC algorithm operation is show in Figure 2. Its operating parameters are as follows: The size of the population was set as 40. The number of food (Food Number) was set as 20. The maximum evolution number of the algorithm termination (maxCycle) was set as 7. The limit (Limit) was set as 20. Two objective functions are set as:

For the design of optimal bandwidth, the objective function F1 is the same as it’s in Equation (1), where ${\left|f_{i}\right|}_{RL=-10\;}$ denotes the frequency points when the reflectivity reaches −10 dB.

For the design of optimal reflectivity, the objective function F2 is the same as it’s in Equation (2), where $RL\left(\theta ,f_{i}\right)$ denotesthe reflectivity of all frequency all frequency points in the design band at various angles of incidence.

## 3. Results and Discussions

### 3.1. Complex Permittivity and Complex Permeability of Carbon Nanotubes 

As a typical one-dimensional nanomaterial, carbon nanotubes (CNT) exhibit unusual electromagnetic wave absorption performance. In Figure 3a, the $\varepsilon^{\prime }$ of the sample with a carbon nanotube weight ratio content of 7 wt.% is basically constant, while the $\varepsilon^{\prime }$ of the sample with a nanotube content of 9 wt.% decreased slightly with frequency from 10.0 to 8.7 GHz. The $\varepsilon^{\prime }$ of the sample increased with the increase of carbon nanotubes content. In Figure 3b, the $\varepsilon^{\prime \prime }$ values of each sample increased with the increase of frequency. As the content of carbon nanotubes increases, the ε” of the materials also increases and the ε” of samples with carbon nanotube content of 7 wt.% and 9 wt.% increased from 0.6 and 1.2 to 2.0 and 3.6, respectively. The ε” curve has some maximum values due to various polarizations [21,39,40,41]. At low frequencies, the weaker space charge polarization is dominant, and at high frequencies, the dipole polarization is dominant. Correspondingly, there are some loss peaks on the $\varepsilon^{\prime \prime }$ curve, which are at 3.5, 5.8, 10.3, 12.8, 14.8, 17.1 GHz, respectively. The samples with different contents of carbon nanotubes had little change for the $\mu^{\prime }$ (from 1.0 to 1.2) and the $\mu^{\prime \prime }$ (from −0.1 to 0.1).

### 3.2. Complex Permittivity and Complex Permeability of of Fe@C Nanoparticles

The Fe@C nanoparticles have both the magnetic loss capability and the dielectric loss capability, which can establish a good impedance match and exhibit excellent MA performance. The Fe@C nanoparticles have a shell-core structure with nano-iron of 30 to 100 nm as the core, and a multi-layer graphitized carbon layer of 3 to 5 nm thickness as shell. The graphite carbon layer of Fe@C nanoparticles can both effectively protect the metal core from being oxidized and prevent the nanoparticles from agglomerating. In Figure 4, the ε′ increase slightly with increasing frequency. There are some significant dielectric relaxation peaks on the ε” curve, which are at 6.0, 8.2, 11.0, 13.5, 15.3, and 17.8 GHz, respectively. The interface polarization that occurs at the interface between the iron core and the carbon shell also contributes to dielectric loss of Fe@C.

The $\mu^{\prime }$ decreases monotonically with increasing frequency, and the $\mu^{\prime }$ of the Fe@C/paraffin complex with Fe@C content of 40 and 50 wt.% decreases from 1.53 and 1.41 to 1.11 and 0.99, respectively, as shown in Figure 4c,d. It shows a wide maximum range appears in the imaginary part µ″, which appears around 8.40 GHz. The maximum value of µ″ corresponding to Fe@C content composite is about 0.28 and 0.20, respectively. In Fe@C nanoparticles, the direct contact between metallic iron nanoparticles is negligible due to the presence of a carbon shell, dipole interaction is the main mechanism [42,43], and the magnetic loss is mainly due to natural resonance.

### 3.3. Complex Permittivity and Complex Permeability of Ni@C Nanoparticles

The Ni@C nanoparticles are made of nano-magnetic metal iron as the core and coated with multiple graphitized carbon layers. From electromagnetic parameter curves as shown in Figure 5, it can be seen with the frequency increases, the $\varepsilon^{\prime }$ of all samples shows a downward trend, and the $\varepsilon^{\prime \prime }$ shows an upward trend. As the content of Ni@C nanoparticles increased, the $\varepsilon^{\prime }$ decreased from 4.8, 7.6, 9.7 to 4.5, 5.9, 6.0, and the $\varepsilon^{\prime \prime }$ increased from 0.5, 1.2, 1.8 to 0.6, 2.9, 3.9, respectively. Additionally, there are some relaxation peaks on the $\varepsilon^{\prime \prime }$ curve, which are attributed to dielectric relaxation in carbon shell and ferromagnetic metal core. With the increase of the content of Ni@C, the values of $\mu^{\prime }$ and $\mu^{\prime \prime }$ decreased from 2 to 18 GHz. The magnetic loss mechanism of Ni@C nanoparticles is also mainly a natural resonance [25].

### 3.4. Complex Permittivity and Complex Permeability of Fe Powder

The micron iron powder has high electromagnetic parameters, so the addition of micron iron powder may contribute to the microwave absorption of the composite. In Figure 6a, we can see that increasing the ball milling time will cause the sample to have a smaller ε′ because of smaller particle size. The ε′ decreases slightly with increasing frequency for all samples. Increasing the ball milling time will reduce the average particle size of the iron powder, so that the polarization of the iron powder will weaken and ε′ will decrease. Meanwhile, in Figure 6b, all samples have dielectric loss peaks at 6, 8.1, 10, 12.8, 15.3 and 16.5 GHz for ε” curve. The $\mu^{\prime }$ (Figure 6c) of all samples is between 2.0–1.0, showing a decreasing trend. Since it is a ferromagnetic material, the permeability μ is higher than that of carbon nanotubes and carbon-coated metal nanoparticles. The μ″ of all samples is between 0.25 and 0.6, and loss peaks appear at about 3.5, 9.0, 11.5, and 16.2 GHz (Figure 6d).

### 3.5. Analysis of GA Algorithm Optimization Results

For two-layers absorbing materials, F1 is used as the objective function of optimizing bandwidth. The first nine optimization results of the program operation (the same below) are showed in Table 2. The obtained convergence time is 2.1 s. It converges in the 17th generation. The optimal result is obtained after the 17th generation: the first layer of material is the number of materials, 9, and the thickness is 0.5 mm; the second layer is 4 and the thickness is 2.5 mm. The optimal bandwidth is 8.12 GHz and the corresponding reflectivity is −42.45 dB. 

In the optimization of the GA algorithm, F2 is used as the objective function of optimizing reflectivity. Table 3 shows the optimization results of the program. The obtained convergence time is 3.9 s. It converges in the 31st generation. The optimal results are obtained after the 31st generation: the first layer material is the number of materials 0 and the thickness is 1.4 mm. The second layer material is 8 and the thickness is 1.6 mm. The optimal reflectivity is −53.41 dB and corresponding bandwidth is 1.64 GHz. The optimized reflectivity curves are showed in Figure 7a.

For three-layers absorbing materials, F1 is used as the objective function of optimizing bandwidth. The first nine optimization results of the program operation are as shown in Table 4. The bandwidth convergence time of the three-layer material optimized by GA is 24.1 s. It converges in the 134th generation. The optimal results are obtained after the 134th generation: the first layer material is 9 and the thickness is 0.6 mm. The second layer material is 4 and the thickness is 1.9 mm. The third layer material is 7 and the thickness is 0.5 mm. The optimal bandwidth is 10.6 GHz and corresponding reflectivity is −20.35 dB.

For three-layers absorbing materials, F2 is used as the objective function of optimizing reflectivity. The first nine optimization results are listed in Table 5. The reflectivity convergence time is 29.6 s. It converges in generation 153. The optimal results are obtained after generation 153: The first layer material is 5 and the thickness is 0.7 mm. The second layer material is 4 and the thickness is 1.0 mm. The third layer material is 7 and the thickness is 1.3 mm. The optimal reflectivity is −84.86 dB and corresponding optimal bandwidth is 3.6 GHz. The optimized reflectivity curves are shown in Figure 7b.

### 3.6. Analysis of ABC Algorithm Optimization Results

For two-layers absorbing materials, F1 is used as the objective function of optimizing bandwidth. The first nine optimization results of the program operation are shown in Table 6. The convergence time is 0.8 s. It converges in the 7th generation. The optimal result is obtained after the 7th generation: the first layer material is 9 with a thickness of 0.5 mm. The second layer material is 4 with a thickness of 2.5 mm. The optimal bandwidth is 8.12 GHz and corresponding reflectivity is −42.45 dB. In the optimization of the ABC algorithm, F2 is used as the objective function of optimizing reflectivity. The optimization results are shown in Table 7. The convergence time obtained is 1.54 s. It converges in the 14th generation. The optimal results are obtained after the 14th generation: the first layer material is 0 and the thickness is 1.4 mm. The second layer material is 8 and the thickness is 1.6 mm. The optimal reflectivity is −53.41 dB, and corresponding bandwidth is 1.64 GHz. The optimized curves can be seen in Figure 8a.

For three-layers absorbing materials, F1 is used as the objective function of optimizing bandwidth. The results can be seen in Table 8. The bandwidth convergence time is 10.6 s. It converges in the 69th generation, and the best results are obtained after the 69th generation: the first layer of material is 9 and the thickness is 0.6 mm. The second layer of material is 4 and the thickness is 1.9 mm. The third layer material is 7 and the thickness is 0.5 mm. The optimal bandwidth is 10.6 GHz and corresponding reflectivity is −20.35 dB.

F2 is used as the objective function of optimizing reflectivity. Table 9 shows the first nine optimization results. The convergence time obtained is 14.4 s. It converges in the 88th generation, and the best results are obtained after the 88th generation: the first layer is 5 and the thickness is 0.7 mm. The second layer is 4 and the thickness is 1.0 mm. The third layer is 7 and the thickness is 1.3 mm. The optimal reflectivity is −84.85 dB and corresponding bandwidth is 3.6 GHz. The optimized curves are shown in Figure 8b.

### 3.7. Comparison of Two Algorithms

It can be seen from Figure 9 that for the bandwidth as the objective function for the two-layer material, the GA reaches convergence in the 17th generation and takes 2.1 s, and the ABC algorithm reaches convergence in the 7th generation and takes 0.8 s. Compared with GA, the ABC algorithm can get the optimal solution in the shortest time and the highest efficiency. For the optimal reflectivity as the objective function for the two-layer material, the GA algorithm reaches convergence in the 31st generation and takes 3.9 s, and the ABC algorithm reaches convergence in the 14th generation and takes 1.54 s. The ABC algorithm can obtain the optimal solution in the shortest time and the highest efficiency.

In Figure 10a, the GA reaches convergence in the 134th generation and takes 24.1 s; the ABC algorithm reaches convergence in the 69th generation and takes 10.6 s. Compared with GA, the ABC algorithm can obtain the optimal solution in the shortest time and with the highest efficiency. In Figure 10b, the GA reaches convergence in the 153rd generation and takes 29.6 s. The ABC algorithm reaches convergence in the 88th generation and takes 14.4 s. Therefore, the ABC algorithm can obtain the optimal solution in the shortest time and the highest efficiency. The advantage of ABC algorithm is that global and local search is carried out to avoid falling into the local optimal situation to some extent.

## 4. Conclusions

We selected GA and ABC algorithm to optimize the absorbing properties of 10 kinds of materials. The ABC algorithm is used to optimize the three-layer absorbing material and the optimization result with bandwidth as the objective function is: the first layer material is “Fe-10 h” and the thickness is 0.6 mm; the second layer material is “Ni@C-40%” and the thickness is 1.9 mm; and the third layer material is “Ni@C-70%” and the thickness is 0.5 mm. The optimal bandwidth is 10.6 GHz and corresponding reflectivity is −20.35 dB. The optimization result with the reflectivity as the objective function is: the first layer is “Ni@C-50%” and the thickness is 0.7 mm; the second layer is “Ni@C-40%” and the thickness is 1.0 mm, and the third layer material is “Ni@C-70%” and the thickness is 1.3 mm. The optimal reflectivity is −84.86 dB and corresponding bandwidth is 3.6 GHz. After optimization, obtained reflectivity and bandwidth are better than that of the single material in the same thickness. This shows that the absorbing performance (bandwidth and reflectivity) of these new absorbing materials can be optimized through the algorithm in the multi-layer absorbing coating, especially the application of ABC algorithm, has not yet seen in the same research.

Comparing the optimization application of GA and ABC algorithm in multi-layer absorbing coating. For the optimal bandwidth, the GA reaches convergence in the 17th generation and takes 2.1 s and the ABC algorithm reaches convergence in the 7th generation and takes 0.8 s. For the optimal reflectivity, the GA reaches convergence in the 31st generation and takes 3.9 s and the ABC algorithm reaches convergence in the 14th generation and takes 1.54 s. Compared with the GA, the ABC algorithm application can obtain the optimal solution in a shorter time and higher efficiency. When different optimization algorithms are used to optimize the three-layer material, for optimal bandwidth as the objective function, the GA reaches convergence in the 134th generation and takes 24.1 s and the ABC algorithm reaches convergence in the 69th generation and takes 10.6 s. For optimal reflectivity as the objective function, the GA reached convergence in the 153rd generation and takes 29.6 s. The ABC algorithm reaches convergence in the 88th generation and takes 14.4 s. It is found that the ABC algorithm has less convergence time and running time than GA, and the algorithm efficiency is higher. 

## Figures and Tables

**Figure 1 nanomaterials-11-01951-f001:**
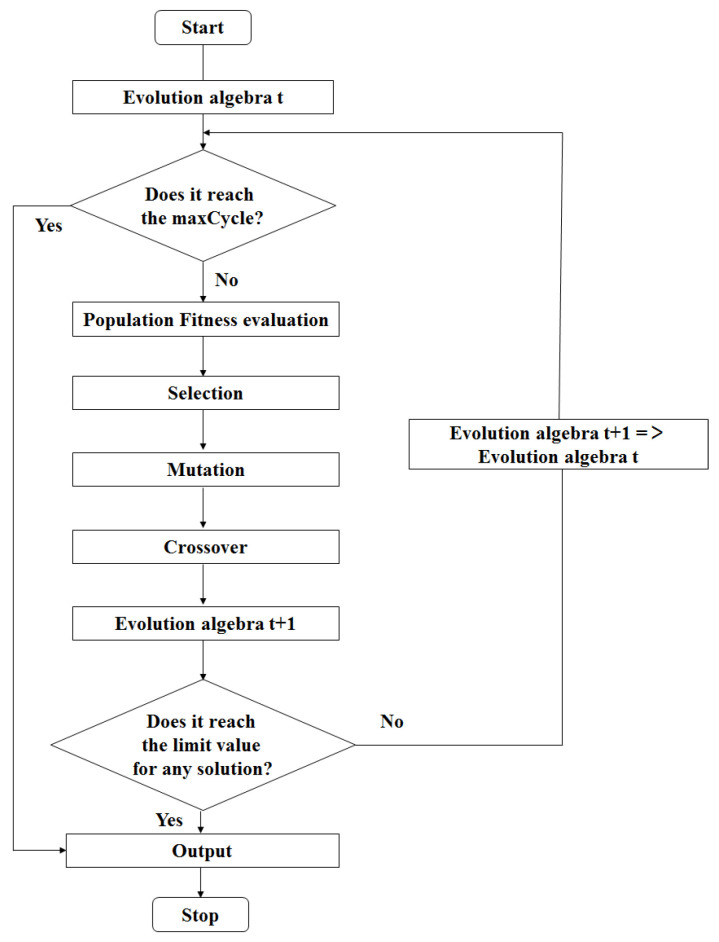
Flowchart of GA algorithm.

**Figure 2 nanomaterials-11-01951-f002:**
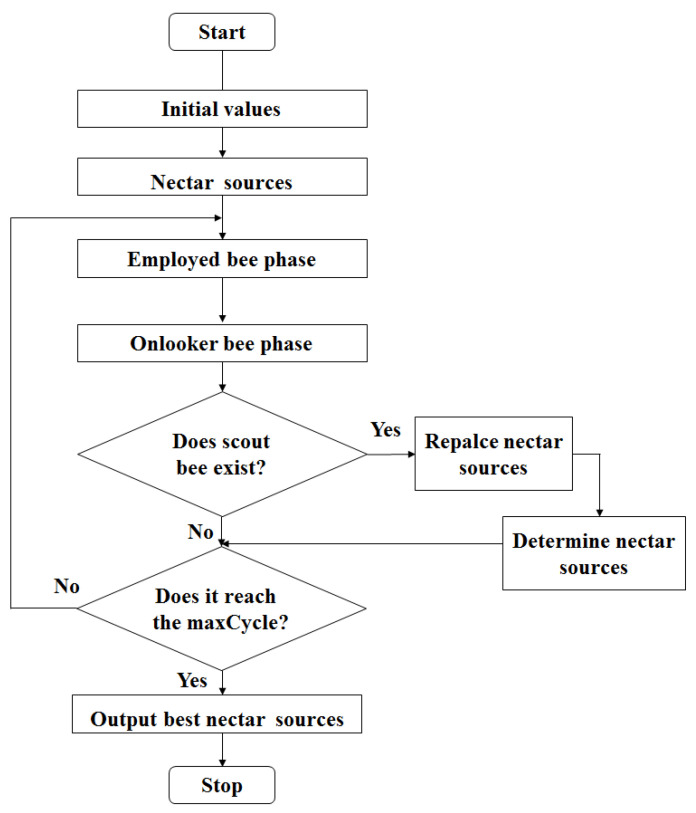
Flowchart of ABC algorithm.

**Figure 3 nanomaterials-11-01951-f003:**
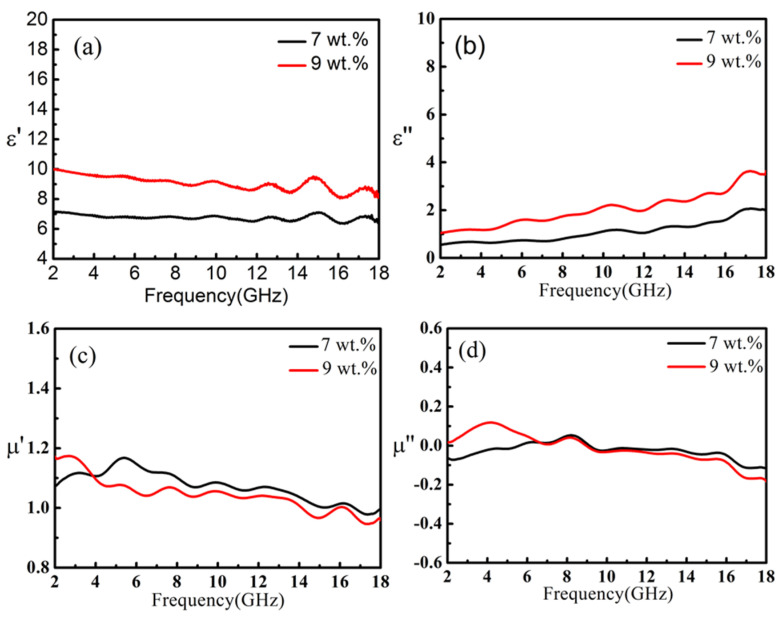
The $\varepsilon^{\prime }$ (**a**), $\varepsilon^{\prime \prime }$ (**b**), $\mu^{\prime }$ (**c**) and $\mu^{\prime \prime }$ (**d**) of carbon nanotubes/paraffin samples.

**Figure 4 nanomaterials-11-01951-f004:**
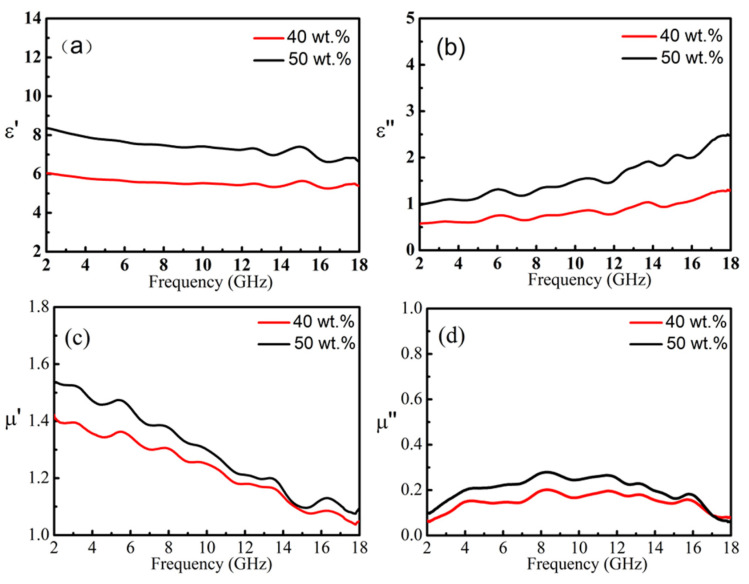
The $\varepsilon^{\prime }$ (**a**), $\varepsilon^{\prime \prime }$ (**b**), $\mu^{\prime }$ (**c**) and $\mu^{\prime \prime }$ (**d**) for Fe@C /paraffin composites containing 40 wt.% and 50 wt.%, respectively.

**Figure 5 nanomaterials-11-01951-f005:**
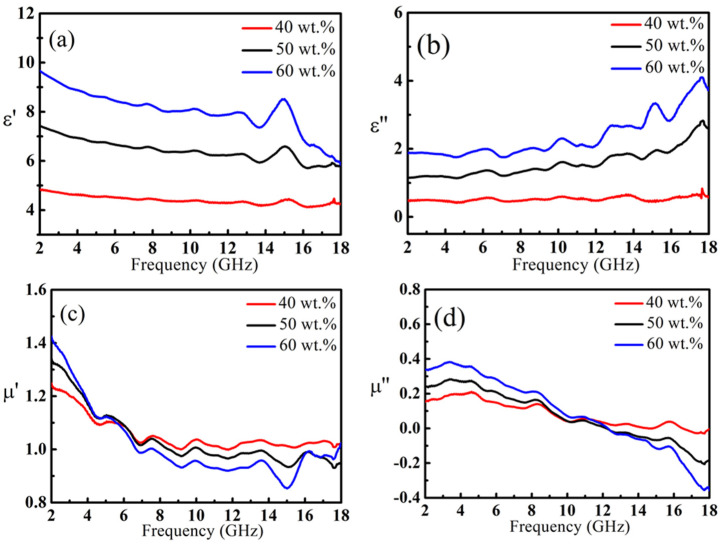
The $\varepsilon^{\prime }$ (**a**),$\varepsilon^{\prime \prime }$ (**b**), $\mu^{\prime }$ (**c**) and $\mu^{\prime \prime }$ (**d**) of Ni@C/paraffin composites.

**Figure 6 nanomaterials-11-01951-f006:**
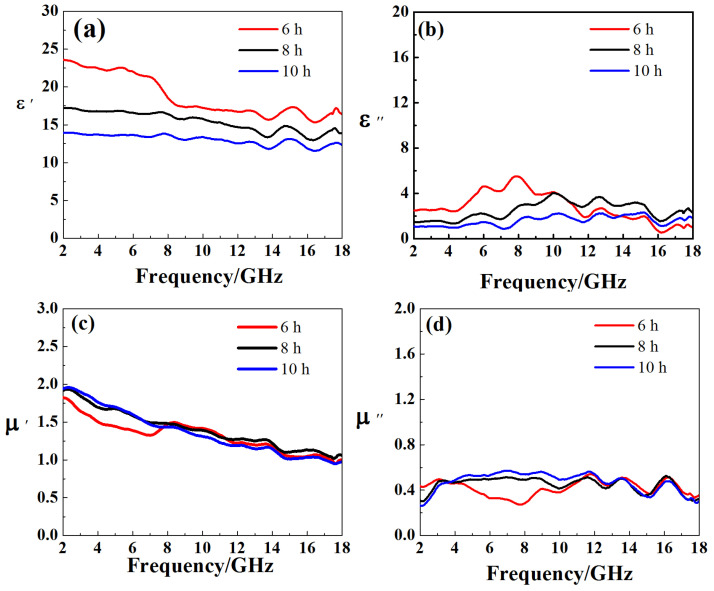
The ε′ (**a**), ε″ (**b**) $\mu^{\prime }$ (**c**) and μ″ (**d**) of samples with different milling time.

**Figure 7 nanomaterials-11-01951-f007:**
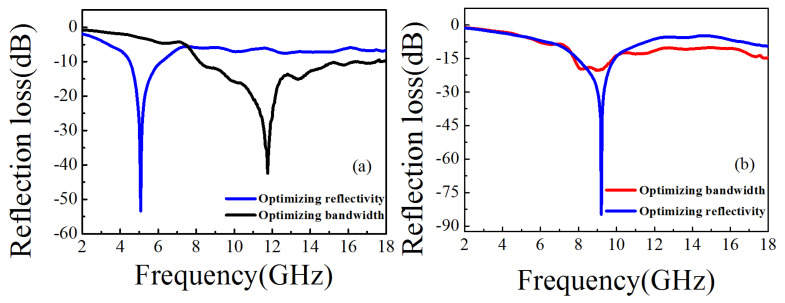
The optimized curves for the two-layer materials (**a**); the optimized curves for the three-layer materials (**b**).

**Figure 8 nanomaterials-11-01951-f008:**
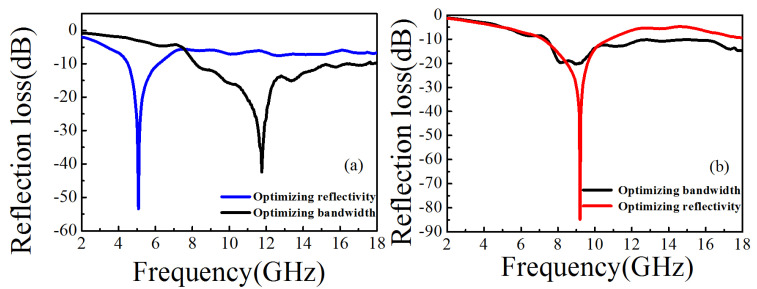
The optimized curves for the two-layer materials (**a**); the optimized curves for the three-layer materials (**b**).

**Figure 9 nanomaterials-11-01951-f009:**
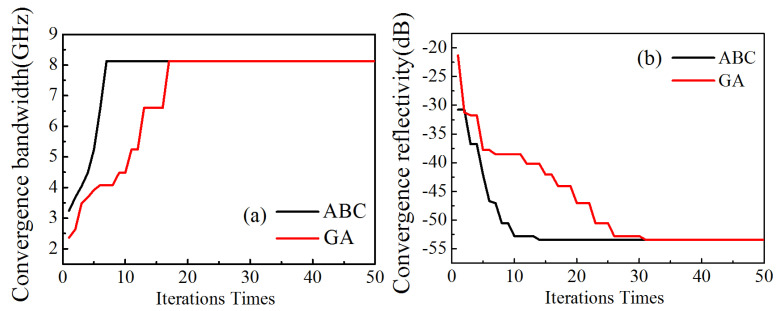
The convergence velocity comparison of two algorithms for the optimal bandwidth (**a**) and reflectivity (**b**) for two-layer materials.

**Figure 10 nanomaterials-11-01951-f010:**
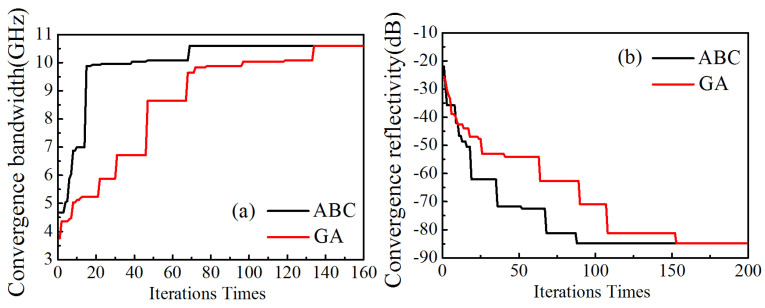
The convergence velocity comparison of two algorithms for the optimal bandwidth (**a**) and optimal reflectivity (**b**) for three-layer materials.

**Table 1 nanomaterials-11-01951-t001:** The number of materials.

Number	Materials
0	CNT-7%
1	CNT-9%
2	Fe@C-40%
3	Fe@C-50%
4	Ni@C-40%
5	Ni@C-50%
6	Ni@C-60%
7	Ni@C-70%
8	Fe-8 h-50%
9	Fe-10 h-50%

**Table 2 nanomaterials-11-01951-t002:** The optimization results using objective function F1 (two-layers) of GA algorithm.

Frist Layer Materials	Second Layer Materials	Material Thickness of the First Layer (mm)	Material Thickness of the Second Layer (mm)	Optimal Bandwidth (GHz)	Reflectivity (dB)
9	4	0.5	2.5	8.12	−42.45
9	4	0.6	2.4	6.60	−21.92
7	2	0.5	2.5	5.24	−31.47
7	4	0.5	2.5	5.16	−34.71
7	2	0.6	2.4	5.12	−21.90
7	2	0.7	2.3	4.84	−17.77
9	2	0.5	2.5	4.84	−30.90
7	4	0.6	2.4	4.64	−21.25
9	2	0.6	2.4	4.64	−22.30

**Table 3 nanomaterials-11-01951-t003:** The optimization results using objective function F2 (two-layers) of GA algorithm.

Frist Layer Materials	Second Layer Materials	Material Thickness of the First Layer (mm)	Material Thickness of the Second Layer (mm)	Optimal Bandwidth (GHz)	Reflectivity (dB)
0	8	1.4	1.6	1.64	−53.41
5	7	1.8	1.2	3.28	−52.77
5	9	1.3	1.7	2.40	−50.50
3	9	0.8	2.2	1.76	−47.02
4	8	1.8	1.2	2.28	−46.71
5	8	1.9	1.1	2.40	−46.39
9	3	0.6	2.4	3.72	−45.91
3	7	1.6	1.4	3.12	−45.38
6	9	1.2	1.8	2.32	−44.82

**Table 4 nanomaterials-11-01951-t004:** The optimization results using objective function F1 (three-layers) of GA algorithm.

Frist Layer Materials	Second Layer Materials	Third Layer Material	Material Thickness of the First Layer (mm)	Material Thickness of the Second Layer (mm)	Material Thickness of the Second Layer (mm)	Optimal Bandwidth (GHz)	Reflectivity (dB)
9	4	7	0.6	1.9	0.5	10.60	−20.35
7	6	7	0.5	1.9	0.6	10.08	−17.33
9	3	4	0.5	0.6	1.9	10.08	−19.57
9	3	4	0.5	0.5	2.0	10.04	−21.30
9	6	4	0.6	0.5	1.9	10.04	−18.21
9	5	4	0.5	1	1.5	9.96	−19.90
7	6	7	0.5	2	0.5	9.92	−17.38
7	6	7	0.6	1.8	0.6	9.92	−16.71
9	5	4	0.5	0.9	1.6	9.92	−20.99

**Table 5 nanomaterials-11-01951-t005:** The optimization results using objective function F2 (three-layers) of GA algorithm.

Frist Layer Materials	Second Layer Materials	Third Layer Materials	Material Thickness of the First Layer (mm)	Material Thickness of the Second Layer (mm)	Material thickness of The Third Laye (mm)	Optimal Bandwidth (GHz)	Reflectivity (dB)
5	4	7	0.7	1	1.3	3.6	−84.86
2	9	0	0.5	0.9	1.6	2.44	−81.13
6	5	7	1	0.8	1.2	3.08	−72.47
9	3	7	0.8	1.6	0.6	3.20	−71.67
2	9	1	1.2	1.1	0.7	2.20	−70.97
2	8	3	0.9	0.5	1.6	2.76	−70.46
6	9	0	1.2	1.3	0.5	2.48	−69.82
9	2	9	0.6	1.1	1.3	2.32	−67.81
2	7	9	1.4	0.6	1.0	2.56	−66.03

**Table 6 nanomaterials-11-01951-t006:** The optimization results using objective function F1 (two-layers) of ABC algorithm.

Frist Layer Materials	Second Layer Materials	Material Thickness of the First Layer (mm)	Material Thickness of the Second Layer (mm)	Bandwidth (GHz)	Optimal Reflectivity (dB)
9	4	0.5	2.5	8.12	−42.45
9	4	0.6	2.4	6.6	−21.92
7	2	0.5	2.5	5.24	−31.47
7	4	0.5	2.5	5.16	−34.71
7	2	0.6	2.4	5.12	−21.90
7	2	0.7	2.3	4.84	−17.77
9	2	0.5	2.5	4.84	−30.90
7	4	0.6	2.4	4.64	−21.25
9	2	0.6	2.4	4.64	−22.30

**Table 7 nanomaterials-11-01951-t007:** The optimization results using objective function F2 (two-layers) of ABC algorithm.

Frist Layer Materials	Second Layer Materials	Material Thickness of the First Layer (mm)	Material Thickness of the Second Layer (mm)	Bandwidth (GHz)	Optimal Reflectivity (dB)
0	8	1.4	1.6	1.64	−53.41
5	7	1.8	1.2	3.28	−52.77
5	9	1.3	1.7	2.4	−50.50
3	9	0.8	2.2	1.76	−47.02
4	8	1.8	1.2	2.28	−46.71
5	8	1.9	1.1	2.4	−46.39
9	3	0.6	2.4	3.72	−45.91
3	7	1.6	1.4	3.12	−45.38
6	9	1.2	1.8	2.32	−44.82

**Table 8 nanomaterials-11-01951-t008:** The optimization results using objective function F2 (three-layers) of ABC algorithm.

Frist Layer Materials	Second Layer Materials	Third Layer Materials	Material Thickness of the First Layer (mm)	Material Thickness of the Second Layer (mm)	Material Thickness of the Third Layer (mm)	Bandwidth (GHz)	Optimal Reflectivity (dB)
9	4	7	0.6	1.9	0.5	10.6	−20.35
7	6	7	0.5	1.9	0.6	10.08	−17.33
9	3	4	0.5	0.6	1.9	10.08	−19.57
9	3	4	0.5	0.5	2	10.04	−21.30
9	6	4	0.6	0.5	1.9	10.04	−18.21
9	5	4	0.5	1	1.5	9.96	−19.90
7	6	7	0.5	2	0.5	9.92	−17.38
7	6	7	0.6	1.8	0.6	9.92	−16.70
9	5	4	0.5	0.9	1.6	9.92	−20.99

**Table 9 nanomaterials-11-01951-t009:** The optimization results using objective function F2 (three-layers) of ABC algorithm.

Frist Layer Materials	Second Layer Materials	Third Layer Materials	Material Thickness of the First Layer (mm)	Material Thickness of the Second Layer (mm)	Material Thickness of the Third Layer (mm)	Bandwidth (GHz)	Optimal Reflectivity (dB)
5	4	7	0.7	1	1.3	3.6	−84.86
2	9	0	0.5	0.9	1.6	2.44	−81.13
6	5	7	1	0.8	1.2	3.08	−72.47
9	3	7	0.8	1.6	0.6	3.2	−71.67
2	9	1	1.2	1.1	0.7	2.2	−70.97
2	8	3	0.9	0.5	1.6	2.76	−70.47
6	9	0	1.2	1.3	0.5	2.48	−69.82
9	2	9	0.6	1.1	1.3	2.32	−67.82
2	7	9	1.4	0.6	1	2.56	−66.03

## Data Availability

Excluded.

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
