# Peer review of "The Simulation Design of Microwave Absorption Performance for the Multi-Layered Carbon-Based Nanocomposites Using Intelligent Optimization Algorithm"

_nanomaterials, 2021, doi:10.3390/nano11081951_

Round 1

Reviewer 1 Report

Overall, some interesting results on the novel application of an optimization algorithm to multi-layer absorber design. Below are two suggestions to improve the quality of the paper and presentation of the work:

  1. No details are provided on the ABC algorithm beyond the input parameters. Given the paper is showcasing the implementation of such an algorithm and the results are discussed comparing the performance of the algorithm against others (i.e. GA), more details on the algorithm and its operation are required.
  2. The authors claim the oscillations seen in the imaginary permittivity results are on account of material properties (space charge and dipole polarizations for the carbon nanotube samples and dielectric relaxations in the Fe@C and Ni@C samples). If this is the case, further information should be provided on how these losses arise. The oscillations in the experimental data could also result from errors in the measurement methodology (air entrapment in the composite, gaps in electrical couplings etc.).

Further to the above, numerous grammatical errors exist throughout the document. Below are a selection, however I would recommend the authors thoroughly review the document again:

Line 15: “to meet the all requirements” should be changed to “to meet all of these requirements”.

Line 19: “For two algorithms” should be changed to “Two algorithms”.

Line 40: “their alloy” should be “their alloys”.

Line 45: “However, the application of ferromagnetic micro-powder is limited by its nar-44 row absorbing bandwidth, high density, easy oxidation and corrosion resistance”. Should the last item read “and poor corrosion resistance”, assuming you are trying to highlight limiting factors of ferromagnetic materials?

Line 64: the meaning of “They found that the CNT growled” is unclear? Should this read “They found that the CNT growed”?

Line 85: “we can also easy to design a high absorption material within a certain amount of bandwidth” should read “we can also easily design a high absorption material within a certain amount of bandwidth”

Line 90-92: “Compared with single layer absorbing coating, multi-layer coatings can flexible design coating (including material and thickness of each layer) to obtain a better performance”. This sentence is not clear to the reader, are you suggesting multi-layer materials have better design flexibility as more variables can be adjusted (i.e. material type and thickness of each layer)?

Line 108: the author states: “the reflectivity in the specified frequency interval achieves the best”. Is the suggestion that the reference, Wang [31], provides the best absorption performance currently available, in which case numerical values for comparison would be good. Alternatively, if the author is suggesting that GA is the best optimization algorithm for this application, evidence of this should be provided.

Line 112: “the GA optimization has higher accuracy and larger calculation”, what does the author mean by ‘larger calculation’?

Author Response

Responses to the reviewers

Reviewer 1

Comments and Suggestions for Authors

Overall, some interesting results on the novel application of an optimization algorithm to multi-layer absorber design. Below are two suggestions to improve the quality of the paper and presentation of the work:

  1. No details are provided on the ABC algorithm beyond the input parameters. Given the paper is showcasing the implementation of such an algorithm and the results are discussed comparing the performance of the algorithm against others (i.e. GA), more details on the algorithm and its operation are required.

Response:

Thank you for your suggestions. Considering that the part of algorithm operation details will make the article too long, we had put the operation process of ABC algorithm and GA algorithm in the Supplementary in our previous submission. According to your suggestion, now we put back the operation process of ABC algorithm and GA algorithm in the revised version and mark it more clearly.

  1. The authors claim the oscillations seen in the imaginary permittivity results are on account of material properties (space charge and dipole polarizations for the carbon nanotube samples and dielectric relaxations in the Fe@C and Ni@C samples). If this is the case, further information should be provided on how these losses arise. The oscillations in the experimental data could also result from errors in the measurement methodology (air entrapment in the composite, gaps in electrical couplings etc.).

Response:

We have looked at some literatures about relevant material and found that the curves of the imaginable permittivity are the same with ours. These studies all thought that this is due to various polarizations (space charge and dipole polarizations and dielectric relaxations). We have quoted these literatures in references [36-39], but as far as our knowledge is concerned, we have not been able to give a theoretical explanation for this.

[36]X.G. Liu, Z.Q. Ou, D.Y. Geng, Z. Han, J.J. Jiang, W. Liu, Z.D. Zhang, Influence of a graphite shell on the thermal and electromagnetic characteristics of FeNi nanoparticles, Carbon. 48 (2010) 891–897.

[37]Z. Han, D. Li, H. Wang, X.G. Liu, J. Li, D.Y. Geng, Z.D. Zhang, Broadband electromagnetic-wave absorption by FeCo/C nanocapsules, Appl. Phys. Lett. 95 (2009) 023114.

[38]X. Jian, G. Chen, H. Liu, N. Mahmood, S. Zhu, L. Yin, H. Tang, W. Lv, W. He, K.H.L. Zhang, Q. Zeng, B. Li, X. Li, W. Zhang, X. Wang, Vapor–Dissociation–Solid Growth of Three-Dimensional Graphite-like Capsules with Delicate Morphology and Atomic-level Thickness Control, Crystal Growth & Design. 16 (2016) 5040–5048.

[39]X.G. Liu, Z.Q. Ou, D.Y. Geng, Z. Han, Z.G. Xie, Z.D. Zhang, Enhanced natural resonance and attenuation properties in superparamagnetic graphite-coated FeNi 3 nanocapsules, J. Phys. D: Appl. Phys. 42 (2009) 155004.

Further to the above, numerous grammatical errors exist throughout the document. Below are a selection, however I would recommend the authors thoroughly review the document again:

Thank you for your help and guidance. According to your suggestion, we have made corresponding changes in the revised version.

Line 15: “to meet the all requirements” should be changed to “to meet all of these requirements”.

Response:The sentence has been modified in the revised version.

Line 19: “For two algorithms” should be changed to “Two algorithms”.

Response:The sentence has been modified in the revised version.

Line 40: “their alloy” should be “their alloys”.

Response:The sentence has been modified in the revised version.

Line 45: “However, the application of ferromagnetic micro-powder is limited by its nar-44 row absorbing bandwidth, high density, easy oxidation and corrosion resistance”. Should the last item read “and poor corrosion resistance”, assuming you are trying to highlight limiting factors of ferromagnetic materials?

Response:The sentence has been modified in the revised version.

Line 64: the meaning of “They found that the CNT growled” is unclear? Should this read “They found that the CNT growed”?

Response:The sentence has been modified in the revised version.

Line 85: “we can also easy to design a high absorption material within a certain amount of bandwidth” should read “we can also easily design a high absorption material within a certain amount of bandwidth”

Response:The sentence has been modified in the revised version.

Line 90-92: “Compared with single layer absorbing coating, multi-layer coatings can flexible design coating (including material and thickness of each layer) to obtain a better performance”. This sentence is not clear to the reader, are you suggesting multi-layer materials have better design flexibility as more variables can be adjusted (i.e. material type and thickness of each layer)?

Response:I'm sorry I didn't write it clearly, but what I meant was:. “Compared with single layer absorbing coating, multi-layer coatings can obtain better performance by adjusting material type and thickness of each layer”. The sentence has been modified in the revised version.

Line 108: the author states: “the reflectivity in the specified frequency interval achieves the best”. Is the suggestion that the reference, Wang [31], provides the best absorption performance currently available, in which case numerical values for comparison would be good. Alternatively, if the author is suggesting that GA is the best optimization algorithm for this application, evidence of this should be provided.

Response:I'm sorry I didn't write it clearly, but what I meant was:. Wang [31] optimized the performance of absorbing coatings by using GA algorithm, and the reflectivity in the specified frequency interval achieves the best comparing no optimization. The sentence has been modified in the revised version.

Line 112: “the GA optimization has higher accuracy and larger calculation”, what does the author mean by ‘larger calculation’?

Response:I'm sorry I didn't write it clearly, but what I meant was: “vast calculating amount”. The sentence has been changed as “the GA optimization has higher accuracy but a vast calculating amount”.

Reviewer 2 Report

The paper is of interest to the audience for the journal. But my feeling is that the focus of the contribution is more about methods and optimization algorithms rather than nano-materials. I am not sure the output of the paper can be directly used from material engineering.

 I have the following major comments:

1/ It is not completely clear why the authors address the frequency band 8-12 GHz. What are the corresponding practical applications involving shielding? The authors should better motivate their work.

2/ Genetic algorithms have already been widely used for multilayered shielding for a long time. See reference below. The authors have to better describe the state of art about optimization of multilayered shields.

Heeralal Gargama, Sanjay Kumar Chaturvedi, and Awalendra K. Thakur, "Design and Optimization of Multilayered Electromagnetic Shield Using a Real-Coded Genetic Algorithm," Progress In Electromagnetics Research B, Vol. 39, 241-266, 2012.  

Oktem, M. H. and B. Saka, "Design of multilayered cylindrical shields using a genetic algorithm," IEEE Transactions on Electromagnetic Compatability, Vol. 43, No. 2, 170-176, 2001.

Jiang, L. Y., X. Y. Li, and J. Zhang, "Design of high performance multilayer microwave absorbers using fast pareto genetic algorithm," Sci. China Ser. E.-Tech. Sci., Vol. 52, No. 9, 2749-2757, 2009.

Dib, N., M. Asi, and A. Sabbah, "On the optimal design of multilayer microwave absorbers," Progress In Electromagnetics Research C, Vol. 13, 171-185, 2010.

3/ The authors should better describe the process of fabrication of materials and experimental procedures. Are all the materials designed and fabricated by the authors?

Author Response

Reviewer 2

Comments and Suggestions for Authors

The paper is of interest to the audience for the journal. But my feeling is that the focus of the contribution is more about methods and optimization algorithms rather than nano-materials. I am not sure the output of the paper can be directly used from material engineering.

 I have the following major comments:

1/ It is not completely clear why the authors address the frequency band 8-12 GHz. What are the corresponding practical applications involving shielding? The authors should better motivate their work.

Response:Microwave absorbing coatings were first used in military stealth technology. Stealth technology can effectively improve the survival and penetration ability of weapons and equipment, showing great power in modern war. Stealth fighter is coated with a layer or multi-layer absorbing material in the fuselage for avoiding radar tracking. Microwave absorbing coatings are not only widely used in military fields, but also increasingly important in civil fields such as electromagnetic compatibility and microwave radiation protection. Microwave equipment, communication transmitting stations, power transmission and transformation equipment and mobile phones all have electromagnetic radiation. Electromagnetic interference produced by electromagnetic radiation not only affects the realization of high performance for electronic products, but also causes long-term or short-term harm to human body. Therefore, it is possible to use microwave absorption coatings for microwave radiation protection. The frequency band of microwave is mostly in the 2-18 GHz. This is also the frequency range commonly studied for absorbing materials, of course in some special cases there are higher frequency. These sentences also have been added in the introduction of the revised version.

2/ Genetic algorithms have already been widely used for multilayered shielding for a long time. See reference below. The authors have to better describe the state of art about optimization of multilayered shields.

Heeralal Gargama, Sanjay Kumar Chaturvedi, and Awalendra K. Thakur, "Design and Optimization of Multilayered Electromagnetic Shield Using a Real-Coded Genetic Algorithm," Progress In Electromagnetics Research B, Vol. 39, 241-266, 2012.  

Oktem, M. H. and B. Saka, "Design of multilayered cylindrical shields using a genetic algorithm," IEEE Transactions on Electromagnetic Compatability, Vol. 43, No. 2, 170-176, 2001.

Jiang, L. Y., X. Y. Li, and J. Zhang, "Design of high performance multilayer microwave absorbers using fast pareto genetic algorithm," Sci. China Ser. E.-Tech. Sci., Vol. 52, No. 9, 2749-2757, 2009.

Dib, N., M. Asi, and A. Sabbah, "On the optimal design of multilayer microwave absorbers," Progress In Electromagnetics Research C, Vol. 13, 171-185, 2010.

Response:Thank you for your help and suggestion. Indeed, genetic algorithm has been applied to the study of microwave absorption and electromagnetic shielding for many years. This paper mainly involves the field of microwave absorption. Two references related to microwave absorption provided by the reviewer (the other two are about electromagnetic shielding) were selected and added to the introduction of the revised version.

3/ The authors should better describe the process of fabrication of materials and experimental procedures. Are all the materials designed and fabricated by the authors?

Response:Thank you for your suggestions. In the materials used in this paper, carbon nanotubes and iron powder are commercially available. Ni@C and Fe@C nanoparticles are prepared by arc discharge method. We have added in detail the process of fabrication of materials and experimental procedures in Experimental 2.1.

Round 2

Reviewer 1 Report

Although further details of the specimen manufacturing methodology (paragraph 2 of Section 2.1), layer materials (Table 1) and optimization results (Tables 2-9) have been included in the manuscript, no details of the ABC algorithm and how it operates appear to have been provided in the manuscript (as requested in previous comments). Given the novel application of this algorithm to this application, details on how the algorithm operate in this context are essential. Further, no references have been provided for the ABC algorithm, there appear to be numerous uses of the algorithm for this application, see e.g.: (https://doi.org/10.1109/DIPED.2018.8543261, https://doi.org/10.1016/j.asoc.2020.106696, https://doi.org/10.1109/TMTT.2019.2919574, https://doi.org/10.26833/ijeg.743661, etc.)

Numerous grammatical errors still exist throughout the paper despite the previous suggestions, for example:

Line 15: Was originally: “to meet the all requirements”, was previously suggested to be changed to: “to meet all of these requirements”, and has been changed to: “meet the all of these requirements”, which is still grammatically incorrect.

Line 95: Was originally: “we can also easy to design a high absorption material within a certain amount of bandwidth”, was previously suggested to be changed to: “we can also easily design a high absorption material within a certain amount of bandwidth”, and has been changed to: “we can also easily to design a high absorption material within a certain amount of bandwidth”, which is still grammatically incorrect.

Author Response

Reviewer 1

Comments and Suggestions for Authors

Although further details of the specimen manufacturing methodology (paragraph 2 of Section 2.1), layer materials (Table 1) and optimization results (Tables 2-9) have been included in the manuscript, no details of the ABC algorithm and how it operates appear to have been provided in the manuscript (as requested in previous comments). Given the novel application of this algorithm to this application, details on how the algorithm operate in this context are essential. Further, no references have been provided for the ABC algorithm, there appear to be numerous uses of the algorithm for this application, see e.g.: (https://doi.org/10.1109/DIPED.2018.8543261, https://doi.org/10.1016/j.asoc.2020.106696, https://doi.org/10.1109/TMTT.2019.2919574, https://doi.org/10.26833/ijeg.743661, etc.)

 Response: Thank you for your advice. According to the comments of reviewers, some references provided about ABC algorithm have been added in the introduction of the manuscript. We also added the flowchart of GA and ABC algorithm operation in the revised manuscript. The operation of ABC algorithm has been widely used in the field of computer. This paper mainly focuses on the innovative application of ABC algorithm in the field of new materials. Therefore, I am sorry that we have not described much about calculation, and the length of this paper has been too long.

Numerous grammatical errors still exist throughout the paper despite the previous suggestions, for example:

Line 15: Was originally: “to meet the all requirements”, was previously suggested to be changed to: “to meet all of these requirements”, and has been changed to: “meet the all of these requirements”, which is still grammatically incorrect.

Line 95: Was originally: “we can also easy to design a high absorption material within a certain amount of bandwidth”, was previously suggested to be changed to: “we can also easily design a high absorption material within a certain amount of bandwidth”, and has been changed to: “we can also easily to design a high absorption material within a certain amount of bandwidth”, which is still grammatically incorrect.

 Response: Thank you for your advice. The sentences above has been modified, and the grammar of the whole text has also been checked and modified.

Reviewer 2 Report

The authors have replied to my comments and improved the overall quality of the paper.

Author Response

Reviewer 2

Comments and Suggestions for Authors

The authors have replied to my comments and improved the overall quality of the paper.

Round 3

Reviewer 1 Report

The authors have incorporated the suggested changes and improved the paper substantially from the first submission.